# Evaluation of the Efficacy of Transglutaminase 1 Gene Delivery by Adeno-Associated Virus into Rat and Pig Skin and Safety of ARCI Gene Therapy

**DOI:** 10.3390/ijms26209976

**Published:** 2025-10-14

**Authors:** Alexey Ponomarev, Ilnur Ganiev, Alexander Aimaletdinov, Milana Mansurova, Angelina Titova, Albert Rizvanov, Valeriya Solovyeva

**Affiliations:** 1Institute of Fundamental Medicine and Biology, Kazan Federal University, Kazan 420008, Russia; alekssponomarev@kpfu.ru (A.P.); amajmaletdinov@kpfu.ru (A.A.); mnchirkova@kpfu.ru (M.M.); angatitova@kpfu.ru (A.T.); vavsoloveva@kpfu.ru (V.S.); 2Division of Medical and Biological Sciences, Tatarstan Academy of Sciences, Kazan 420111, Russia; ilnurgm-vgora@mail.ru

**Keywords:** autosomal recessive congenital ichthyosis, lamellar ichthyosis, transglutaminase 1, adeno-associated virus, gene therapy

## Abstract

Autosomal recessive congenital ichthyosis (ARCI) is a heterogeneous group of inherited keratinization disorders with diffuse skin lesions. It includes lamellar ichthyosis, congenital ichthyosiform erythroderma, and fetal ichthyosis. The common pathognomonic feature is generalized neonatal erythroderma. Lamellar ichthyosis is caused by mutations in the *TGM1* gene encoding transglutaminase 1 (TGM1), leading to a functional deficiency of the enzyme in the epidermis. TGM1 deficiency causes severe keratinization defects and skin barrier impairment (leading to metabolic disorders, growth delay, and bacterial infections), with severe cases risking potentially fatal sepsis. Current therapeutic approaches are only symptomatic. In this study, we analyzed the functionality and safety of an adeno-associated viral vector of serotype 2 encoding *TGM1* (AAV2-TGM1) for gene therapy of lamellar ichthyosis. The functionality of AAV2-TGM1 was confirmed in vitro on HEK293, HaCaT, and SH-SY5Y cells and human primary fibroblasts. A significant increase in TGM1 mRNA, protein levels, and enzymatic activity was shown. The vector was characterized and applied in vivo in rats and pigs. Intradermal injection and topical application resulted in increased protein levels in the skin, as shown by PCR and immunofluorescence. Safety was confirmed by the absence of significant histological, biochemical, and cellular changes. The results demonstrate the promise of AAV2-TGM1 for dermal application in gene therapy of lamellar ichthyosis.

## 1. Introduction

Ichthyoses are a heterogeneous group of dermatological diseases, including both hereditary (autosomal dominant, autosomal recessive and X-linked forms) and acquired conditions, characterized by a common pathogenetic mechanism—a disruption of keratinization processes [1]. Clinically, these diseases are manifested by generalized hyperkeratosis with the formation of large-plate scales, marked xerosis, and impaired skin barrier function. According to modern data, the pathogenesis of ichthyosis may be associated with a wide range of pathological conditions, including malignant neoplasms, autoimmune diseases, metabolic and endocrine disorders, and infectious processes [2]. A special group consists of hereditary forms of ichthyosis caused by monogenic defects that lead to impaired formation of the stratum corneum of the epidermis.

Among hereditary ichthyoses, lamellar ichthyosis (LI) plays a special role as one of the most severe forms of nonsyndromal keratinization disorders characterized by high perinatal lethality (up to 20% of cases). The molecular basis of the disease in most cases involves mutations in the *TGM1* gene encoding the enzyme transglutaminase 1, which plays a key role in the processes of cross-linking of stratum corneum proteins [3]. Deficiency of this enzyme leads to pronounced proliferative hyperkeratosis and impaired formation of corneocytes. Modern studies have revealed that the key pathogenetic mechanism of LI is a complex disorder of lipid metabolism in differentiated keratinocytes, which causes defects in the formation of the extracellular matrix of the stratum corneum and significantly impairs the barrier function of the skin [4].

The clinical picture of LI is characterized by significant polymorphism. The most critical is the neonatal period, when 90% of patients develop the so-called collodion baby phenotype, a specific parchment-like membrane covering the entire body surface [5]. This condition is accompanied by gross disorders of thermoregulation, significant fluid, and electrolyte losses, as well as a high risk of septic complications. Subsequently, the disease manifests by the formation of large lamellar scales of dark brown color with characteristic superficial cracks between them. Associated manifestations include dystrophic changes in the nail plates, alopecia areata, ocular involvement of varying severity (ectropion, keratitis), and possible eversion of mucous membranes [6].

Current therapeutic approaches for LI are exclusively symptomatic and include the use of systemic retinoids, intensive emollient therapy, and prevention of secondary infections [7]. Supportive therapy is of particular importance in the first months of life of patients, when the risk of lethal complications reaches maximum values. However, existing therapies are unable to address the underlying cause of the disease—TGM1 deficiency, which necessitates the development of fundamentally new etiotropic approaches [8].

Gene therapy using viral vectors to deliver functional copies of the TGM1 gene appears to be a promising direction in the treatment of LI. Among various vector systems, adeno-associated viruses (AAVs) demonstrate a number of unique advantages, including low immunogenicity, non-integration into the host genome, and the ability to ensure long-term (up to several years) transgene expression [9]. Of particular interest is the AAV2 serotype, which has shown high efficiency in transducing keratinocytes in experimental models. Studies demonstrating the efficacy of correcting an enzyme deficiency in Sjogren-Larsson syndrome by delivering the fatty aldehyde dehydrogenase (FALDH) gene using AAV vectors have become an important proof of concept [10].

However, existing vector systems require substantial optimization, particularly in terms of improving the efficiency of epidermal stem cell transduction; ensuring stable long-term transgene expression; minimizing potential immune reactions; and developing efficient cutaneous delivery methods [11].

In the present study, a comprehensive evaluation of the efficacy and safety of recombinant AAV2 vector carrying the *TGM1* gene following intradermal injection in two experimental models, rats and pigs, was performed. Special attention was paid to the level of transgene expression in different epidermal layers, the duration of the therapeutic effect, and histological changes in the injection area, as well as potential toxic effects.

## 2. Results

### 2.1. Development of the pAAV-TGM1 Plasmid Vector

The nucleotide sequence of wild-type *TGM1* cDNA contains tandem rare codons that can impede translation or reduce its efficiency. After codon optimization of the wild-type *TGM1* gene cDNA, the codon adaptation index (CAI) improved from 0.81 to 0.93. To improve mRNA stability, the GC content was optimized and regions with high GC content were removed, reducing it from 58.80% to 58.16%. In addition, potential cis-acting sites were eliminated. As a result of codon optimization, the length of the TGM1 amino acid sequence remained unchanged at 817 amino acid residues. The de novo synthesized genetic cassette containing the *TGM1* cDNA was cloned into the pUC57 donor vector. The correct assembly of the pUC57-TGM1 construct was confirmed by restriction analysis. The *TGM1* cDNA cassette was then cloned from the pUC57-TGM1 plasmid into the pAAV-MCS target vector using *Eco*RI/*Bam*HI restriction sites. The resulting pAAV-TGM1 plasmid (Figure 1A) contained the following elements of the AAV vector: CMV promoter/enhancer, β-globin intron, optimized TGM1 sequence, and human growth hormone (hGH) polyadenylation signal. The construct was flanked by AAV2 inverted terminal repeats (ITRs). Plasmid was produced in preparative quantities and, following confirmation of its functionality (Figure 1B–D), was used to assemble recombinant AAV.

The functionality of pAAV-TGM1 was tested in HEK293, HaCaT, SH-SY5Y, and primary human fibroblasts (PHF) cell cultures. Transient transfection was performed to evaluate *TGM1 gene* expression levels and the functional activity of the TGM1-encoded protein. Figure 1B presents TGM1 enzymatic activity data in native and transfected HEK293, HaCaT, and SH-SY5Y cells. Following transfection, significant increases in TGM1 enzyme activity were observed: 5.7-fold in HEK293 cells, 2.15-fold in HaCaT cells, and 7-fold in SH-SY5Y cells (Figure 1D).

Figure 1B illustrates the mRNA level of the *TGM1* gene, as determined by RT-PCR. In all cell lines tested, including PHF, transfection with pAAV-TGM1 resulted in significantly higher transcript levels than in the control groups. Since the primers were specific to the codon-optimized sequence, no expression was observed in the control samples. Statistically significant differences were observed between the pAAV-TGM1 transfected and native groups of HEK293, HaCaT, SH-SY5Y, and PHF cells. Figure 1C shows the results of Western blot analysis of *TGM1* expression in HEK293 cells following transfection with pAAV-TGM1 and pAAV-GFP plasmids. A protein band of approximately 89 kDa, corresponding to the expected size of TGM1, was detected in the pAAV-TGM1 group. No specific band was observed in native cells or in the pAAV-GFP transfected group. β-actin levels were used as a loading control and were comparable between groups. Additionally, functionality of pAAV-TGM1 was assessed using semi-quantitative immunofluorescence analysis and mean fluorescence intensity (MFI) (Section 2.2). The MFI increased 7.75-fold in HEK293 cells, 5.57-fold in SH-SY5Y cells, and 4.13-fold in PHF cells compared to the control.

### 2.2. Analysis of AAV2-TGM1 Functionality In Vitro

The functionality of AAV2-TGM1 was analyzed in HEK293, HaCaT, SH-SY5Y, and PHF cell cultures. For this purpose, genetic modification (transduction) of cells with recombinant AAV2-TGM1 was performed. Immunofluorescence analysis of TGM1 expression in transduced HEK293, SH-SY5Y, PHF, and HaCaT cells showed that the modified cells exhibited a 4–5 fold increase in fluorescence intensity, with HaCaT cells showing endogenous expression. The highest fluorescence intensity was observed in HEK293 cells, which is likely attributable to the high permissiveness of this cell line to viral transduction (Figure 2A,B).

Using RT-PCR, we found that the *TGM1* gene copy number varied from 1 × 10^7^ to 1 × 10^9^ copies per 1 μg of total RNA in samples of transduced HEK293, HaCaT, PHF, and SH-SY5Y cells (Figure 2C). A protein band corresponding to the expected size of TGM1 (89 kDa) was detected in lysates from transduced HEK293 cells using Western blot analysis (Figure 2D).

According to the enzymatic activity assay, TGM1 activity was 16.3-fold higher in transduced HEK293 cells and 2.19-fold higher in HaCaT cells compared to native cells (Figure 2E).

The efficiency of vector transduction and the level of transgene expression are significantly affected by the quality of the preparation batch. Important criteria include the proportion of capsids carrying recombinant DNA and the absence of impurities. Transmission electron microscopy analysis of the obtained AAV2-GFP and AAV2-TGM1 viral vectors revealed a predominance of full (genome-containing) capsids, as evidenced by high electron density in their central region (Figure 3A). The average diameter of the viral particles was 21.021 ± 1.572 nm for AAV2-GFP and 22.453 ± 1.758 nm for AAV2-TGM1. Analysis of vector preparation purity demonstrated that both were free from cellular debris, viral particle aggregates, and protein contaminants.

The transduction efficiency of HEK293, SH-SY5Y, HaCaT, and PHF cells with the AAV2-GFP vector was analyzed using fluorescence microscopy and flow cytometry. The percentage of GFP-positive HaCaT cells was 16%, which was lower than that of HEK293 (88%), SH-SY5Y (95%), and PHF (51%) (Figure 3B). In our study, we used different cell lines to determine the versatility and efficacy of AAV2, which is important for understanding its potential in future therapeutic studies.

Assessment of cytotoxicity in vitro demonstrated that transduction with the both AAV2-TGM1 and AAV2-GFP vectors led to a 7–8% reduction in the viability of HEK293 cells at 7 days post-transduction (Figure 4A). The effects of transfection with pAAV-TGM1 and transduction with AAV2-TGM1 on the cytokine profile of HaCaT and PHF cells were also analyzed (Figure 4B). The plasmid vector pAAV-TGM1 showed moderate effects on the levels of pro-inflammatory (interleukin 1 alpha (IL-1α), monocyte chemoattractant protein 1 (MCP-1)) and regenerative markers (basic fibroblast growth factor (bFGF), hepatocyte growth factor (HGF)), whereas the viral vector resulted in a clear increase in the levels of these markers, especially in fibroblasts, suggesting a potential for provoking a more pronounced inflammatory response. The absence of changes in key markers (interferon gamma-induced protein 10 (IP-10), cutaneous T-cell attracting chemokine (CTACK)) associated with antiviral immunity, and T-lymphocyte migration into the skin is a favorable characteristic for the potential use of the vector in gene therapy. Additional, larger-scale studies are warranted to make specific conclusions.

### 2.3. Functionality and Safety Analysis of AAV2-TGM1 in Rats

In the in vivo evaluation of vector functionality, using RT-PCR showed a 1.6-fold higher transgene expression of *TGM1* in the topical application group compared to the control group. The intradermal injection group also showed higher expression than the control, but this difference did not reach statistical significance. Immunofluorescence analysis confirmed an increase in TGM1 protein expression in the rat epidermis following topical application of AAV2-TGM1, representing a 1.41-fold increase compared to the control group treated with PBS (Figure 5).

Morphological changes in the skin, parenchymatous, and immune system organs of rats after intradermal injection and topical application of AAV2-TGM1 were evaluated. No infiltration of inflammatory cells in the dermis or thickening of the epidermis was observed in the experimental groups (Figure 6A,B).

Evaluation of the parenchymatous organs and the central nervous system (CNS) revealed no pronounced inflammatory infiltrates, necrosis, or dystrophic changes in the experimental and control groups (Figure 7A). However, isolated instances of thickening of the interstitium and interalveolar septa in the lungs were observed in the AAV2-TGM1 injected group (red arrow), as well as localized death of spinal cord neurons in both control and experimental groups. Single samples of liver tissue in the control and experimental groups showed focal fine fatty dystrophy (representative images of these incidental findings are not shown due to their low incidence). Importantly, biodistribution analysis of the AAV2-TGM1 vector demonstrated a 6-fold increase in transgene mRNA levels in the liver and a 3-fold increase in the kidney following topical application, as well as a 4-fold increase in the heart after intradermal injection on day 7. These findings indicate that both applicative and intradermal delivery can result in systemic distribution of the viral vector through the bloodstream. None of the examined organs showed signs of acute toxicity, hemorrhage, or significant tissue degeneration.

To comprehensively assess the safety of the vector, the dynamics of blood biochemistry parameters, leukocyte formula, cellularity, and body weight of animals were analyzed. The concentration of total bilirubin, creatinine, ALT, and AST varied between the groups depending on the method of administration and the observation period, but in most cases remained within the reference ranges. The dynamics of ALT in the experimental and control groups did not have strong variability and was within 45–50 U/L. AST increased sharply by day 7 in all groups; following topical AAV2-GFP application, the level reached a maximum of 128.05 U/L by day 21, while in the topical application AAV2-TGM1 group, the level decreased to 61 U/L, showing more moderate changes. Concentrations of total bilirubin fluctuated, increasing on day 7, decreasing on day 14, and sharply increasing to the highest values on day 21 in both the control and experimental groups. The creatinine concentration in the AAV2-GFP injection group and the AAV2-TGM1 injection group was slightly lower than that in the control, whereas an increase was observed in the topical AAV2-GFP and AAV2-TGM1 groups by day 21 (Figure 8A).

Statistical analysis with a significance threshold of *p* < 0.05 showed that none of the experimental groups (AAV2-GFP injection, AAV2-TGM1 injection, AAV2-GFP application, and AAV2-TGM1 application) exhibited significant changes in the proportions of lymphocytes, monocytes, and granulocytes compared to baseline (day 0) or to the PBS-injected control group at any time point (days 7, 14, and 21) (Figure 8B).

The viability of cells in rat lymph nodes remained high in all groups throughout the observation period, with slight fluctuations; in the control group with PBS injection, the indices varied from 92.4% to 95.7%. In the experimental groups, stable values were also observed, except for a slight decrease in the topical application AAV2-TGM1 group on the 21st day (88.3%). In the thymus, cell viability was decreased relative to the control group, with the most pronounced differences observed in the AAV2-TGM1 and AAV2-GFP injection groups, where viability decreased to 67.7% and 75.2%, respectively. In contrast, higher viability (80% to 91.6%) was maintained in the control group and the topical application AAV2-GFP and AAV2-TGM1 groups. In the bone marrow, cell viability in all groups was slightly lower than in the lymph nodes and thymus ranging from 66.1% to 81.4%. The most pronounced decrease was observed in the AAV2-TGM1 injection group on day 21. Overall, topical application of AAV2-GFP and AAV2-TGM1 was associated with fewer alterations in cell viability compared to intradermal injection (Figure 9A).

In the control group with PBS injection and in the groups with AAV2-GFP and AAV2-TGM1 injection, the body weight of rats remained stable or increased during the experiment, indicating that there was no pronounced adverse effect on the animals’ general condition. In the topical application groups (AAV2-GFP and AAV2-TGM1), a tendency toward decreased body weight was observed by day 21, suggesting a potential adverse effect of the scarification procedure used for the topical applications (Figure 9B).

### 2.4. Functionality and Safety Analysis of AAV2-TGM1 in Pigs

RT-PCR analysis detected TGM1 transgene mRNA in pig skin at 1, 3, and 7 days after intradermal injection of AAV2-TGM1, showing 4.5-fold higher expression at day 1 and 2.64-fold higher expression at day 3 compared to day 7 (Figure 10A). The increased recombinant RNA level correlated with the results of immunofluorescence analysis, which demonstrated enhanced *TGM1* gene expression and protein level in the epidermis compared with the control group receiving PBS (Figure 10B). The group of pigs with AAV2-TGM1 injection showed a significant 4.21-fold increase in fluorescence intensity compared to the control group.

Morphological changes in the skin, parenchymatous organs, and central nervous system of pigs after AAV2-TGM1 injection were evaluated. Examination of skin biopsies from experimental animals revealed no inflammatory infiltration, dystrophic, or necrotic changes. Histopathological analysis showed no signs of acute toxicity, such as significant inflammatory infiltrates, necrosis, or dystrophic changes in the dermis. The cellular structure of the epidermis and intercellular matrix were preserved, and the tissue architecture was intact. The vascular network showed no signs of damage or pathological neovascularization (Figure 10C,D).

Histological analysis of parenchymatous organs (liver, kidneys, heart, lungs, spleen), as well as the brain and spinal cord of pigs revealed no significant toxic effects. No inflammatory infiltrates, necrotic and dystrophic changes, and disturbance of architectonics were observed in the examined tissues. The structure of hepatocytes, renal tubules, cardiomyocytes, and alveolar epithelium was preserved, without signs of degeneration or fibrosis. No demyelination, neurodegeneration, or microglial activation was detected in the brain and spinal cord. Compared with the control group, the other organs appeared normal, with no obvious pathological changes (Figure 11A). Based on the histopathological data, intradermal injection of AAV2-TGM1 did not significantly affect the morphology of the CNS or parenchymal organs. This finding is consistent with the biodistribution analysis, which showed no increase in transgene mRNA levels in the organs of pigs after intradermal administration of AAV2-TGM1 compared to the control group (Figure 11C).

Blood biochemical parameters showed transient changes during the experiment: on day 7, levels of AST, total bilirubin, and creatinine were elevated, but these parameters returned to reference ranges by day 28. In contrast, ALT levels remained within the normal range initially but were elevated by the end of the study. These results indicate that intradermal injection of AAV2-TGM1 had no significant lasting effects on blood biochemical parameters by day 28 (Figure 11B).

## 3. Discussion

Currently, gene therapy for skin diseases is primarily being investigated in preclinical studies. In fact, out of 1052 clinical trials involving gene therapy, only 23 focus on skin diseases [11]. The skin is an attractive target for gene therapy for several reasons. The epidermis is frequently affected by monogenic skin diseases and can be readily modeled in vitro or manipulated ex vivo. Furthermore, the effects of therapy can be easily monitored. Additionally, current advances in cell culture techniques have made it possible to expand keratinocytes, including keratinocyte stem cells (KSCs), in vitro [12].

Direct delivery of therapeutic vectors to the skin via topical application or intravenous/intradermal injection is an attractive strategy for treating genodermatoses [13,14]. Several local gene therapies have been developed using modified herpes simplex virus 1 (HSV-1) vector carrying wild-type genes, such as *COL7A1* for recessive dystrophic epidermolysis bullosa or *TGM1* for autosomal recessive congenital ichthyosis [15]. The results of a phase 1 gene therapy with local delivery were promising, although transgene expression in the skin was transient due to the non-integrating nature of HSV-1, necessitating repeated vector injections [12].

In our study, we performed a comprehensive evaluation of the functionality and safety of recombinant AAV2-TGM1. Initially, we developed and characterized the plasmid vector pAAV-TGM1 and demonstrated its functionality in human cells. Transfection of various cell lines with pAAV-TGM1 resulted in successful upregulation of TGM1 mRNA and protein expression and increased enzymatic activity. However, a 67% higher level of enzymatic activity was observed in HEK293 cells compared to HaCaT keratinocytes, likely due to the lower transfection efficiency of the latter. Previous studies have reported that transduction of primary keratinocytes with AAV2 is less than 5% efficient. It has been shown that while primary keratinocytes express AAV2 internalization receptors αvβ5 and α5β1 integrins, they lack heparan sulfate proteoglycan (HSPG), which serves as the primary attachment receptor [16].

After confirming the functionality of pAAV-TGM1, recombinant AAV2-TGM1 was produced based on this plasmid, and its functionality and safety were analyzed both in vitro and in both small and large animal models.

We evaluated the functionality of AAV2-TGM1 in vitro by immunofluorescence analysis of TGM1 expression in epidermal and control cell lines. Endogenous TGM1 expression was observed in HaCaT cells, consistent with data from the Protein Atlas (https://www.proteinatlas.org) (accessed on 12 October 2025). Western blot analysis confirmed TGM1 expression in AAV2-TGM1–modified HEK293 cells, which was also confirmed at the mRNA level by RT-PCR. The use of antibodies allowed to estimate the protein level only relative to the control group, taking into account the level of endogenous expression. In contrast, the use of primers specific to the codon-optimized sequence enabled the specific detection and quantification of recombinant mRNA.

In HEK293 cells after transfection with pAAV-TGM1 plasmid, TGM1 enzymatic activity increased 5.7-fold, and in HaCaT cells, it increased 2.15-fold compared to native cells. Similar results were obtained following viral transduction: TGM1 activity increased 16.3-fold in HEK293 cells and 2.19-fold in HaCaT cells compared to the control. Importantly, the increase in enzyme activity was accompanied by an increase in TGM1 mRNA and protein expression, as confirmed by RT-PCR and Western blot analysis. These findings suggest that even a temporary restoration of TGM1 enzymatic activity could significantly improve skin barrier function, which is supported by replacement therapy studies in animal models of ichthyosis [17,18].

The proportion of full capsids was approximately 80% in both AAV2-TGM1 and AAV2-GFP samples, and the average size of the viral particles was 21.5 nm. Analysis of transduction efficiency in the tested cell lines (HEK293, HaCaT, PHF, and SH-SY5Y) revealed a low transduction level in HaCaT cells, which was confirmed both visually and by flow cytometry. This is likely attributable to the dense adhesion and differentiated state of these keratinocytes, accompanied by keratinization-related thickening of the cellular envelope.

Recombinant AAVs are the leading vectors for a wide range of gene therapy applications in humans due to their favorable safety profile in clinical trials, such as their ability to transduce both dividing and non-dividing cells, low cytotoxicity, and capacity for long-term transgene expression [19,20]. Multiple AAV serotypes from humans and non-human primates have been identified, each with distinct tissue tropism and transduction efficiency profiles. AAVs can efficiently transduce many tissues, including the liver, lung, eye, and skeletal muscle, as their tissue specificity depends on both capsid properties and the route of administration [21].

It was shown that transduction with both AAV2-TGM1 and AAV2-GFP reduced the viability of HEK293 cells by 7–8% by 7 days. The effects of transfection with pAAV-TGM1 and transduction with AAV2-TGM1 on the cytokine profile of HaCaT and PHF cells were also evaluated. The plasmid vector pAAV-TGM1 showed moderate effects on pro-inflammatory (IL-1α, MCP-1) and regenerative markers (bFGF, HGF), whereas AAV2-TGM1 resulted in a stronger induction of these cytokines, especially in PHFs, suggesting a potential for more pronounced inflammation. The absence of changes in key markers (IP10, CTACK) associated with antiviral immunity and T-lymphocyte migration into the skin is a favorable factor for the therapeutic use of the vector in gene therapy [22,23]. Definitely, additional larger-scale studies are necessary for more specific conclusions.

Another research group has also explored the potential of gene therapy for TGM1 deficiency by developing a vector designed to deliver a functional human *TGM1* gene directly into the skin [15]. This vector is based on HSV-1, which has a natural tropism for the epidermis and can more efficiently penetrate skin cells than other viral vectors. The vector was analyzed in vitro using patient-derived keratinocytes and in vivo in mice using a skin permeabilization model involving tape stripping or acetone treatment [24]. Administration of HSV-1-TGM1 at a dose of 1.07 × 10^9^ plaque-forming units (PFUs) was shown increase *TGM1* expression in mouse skin without adverse effects [15].

Although AAV does not have natural mechanisms for penetrating the stratum corneum, it can still transduce skin cells. Physical damage or inflammation can compromise this barrier, facilitating access to keratinocytes in the basal layer and dermal fibroblasts [25]. Furthermore, certain serotypes and engineered capsids (e.g., AAV5, AAV8, AAV-DJ) demonstrate enhanced transduction efficiency in skin cells [16], and high local concentrations achieved through intradermal injection or scarification can further enhance this effect.

The homozygous deletion of the *TGM1* gene in mice results in neonatal lethality [26], making a complete knockout model unsuitable for preclinical studies. We have performed functional studies and evaluated the acute toxicity of AAV2-TGM1 in small (rat) and large (pig) animal models. Chronic toxicity was not assessed due to the development of an immune response against the vector upon repeated administration. Neutralizing antibodies to the capsid and viral genome resulting from adaptive humoral immunity significantly reduce the efficacy of the viral vector without the use of immunosuppression [27].

*TGM1* gene expression in the skin was confirmed in both rats and pigs, indicating the functionality of recombinant AAV2-TGM1 in vivo. Following topical application in rats, a 1.41-fold increase in TGM1 fluorescence intensity was observed, while intradermal injection in pigs resulted in a 4.21-fold increase compared to controls. The observed *TGM1* expression levels were higher than those reported by Freedman et al. [15], where local injection of HSV-1-TGM1 in mice caused a significant increase in protein levels in the absence of toxicity. However, this difference may be attributed to methodological variations and the use of different animal models (mice and pigs). In our study, *TGM1* expression in pigs decreased by 7 days, which is consistent with Jayarajan et al., who described transient expression upon non-integrating vector delivery to the skin [12]. When evaluating the in vivo efficacy of the viral vector in this study, a number of limitations related to quantification have been identified. First, detection following both intradermal and topical application is challenging with standard histological methods. Second, there is cross-reactivity at the protein level with endogenous TGM1. Third, considerable intragroup variability was observed, likely due to factors such as skin heterogeneity, thickness, host immune responses, and even animal sex [28].

Biochemical blood analysis showed a moderate and reversible increase in bilirubin and creatinine in rats, mainly on day 21, which may indicate a transient burden on the liver and kidneys. Similar changes were observed in pigs on day 7, but the values returned to baseline by day 28. These fluctuations fall within the range of expected physiological variation, consistent with observations from other preclinical gene therapy studies [29,30]. Analysis of immune organ cellularity in rats showed a moderate decrease in thymic cell viability in the injection groups, aligning with reports of a localized stress response following AAV administration [31]. The absence of significant changes in bone marrow and lymph nodes, however, suggests minimal systemic immunotoxicity.

The literature describes isolated cases of cutaneous changes following AAV vector administration, including hyperkeratosis and activation of proliferative processes during wound healing, as well as dose-dependent perivascular inflammation and cellular infiltration at high viral titers [17]. However, other studies using comparable AAV doses have reported no significant pathological changes in skin architecture. Consistent with the latter findings, our investigation in both rat and porcine models revealed no histological abnormalities or evidence of inflammatory reactions compared to untreated controls. This finding is consistent with the results of Aufenvenne et al. [17], where topical application of TGM1 using liposomes in humanized mouse model also did not induce structural abnormalities in the skin. The preservation of normal dermal and epidermal architecture, along with the absence of pathological neovascularization, demonstrates the safety of both topical application and intradermal injection routes [17].

Histopathological evaluation of internal organs and the CNS in both rats and pigs showed no significant differences between experimental and control animals. The absence of degenerative changes, inflammation, or neurotoxicity is consistent with observations indicating a favorable safety profile of AAV vectors. The findings add to the existing literature, confirming the promise of AAV2-TGM1 as a platform for gene therapy of cutaneous monogenic diseases with low risk of systemic toxicity, at appropriate doses [32].

In conclusion, AAV2-TGM1 represents a promising candidate for restoring skin barrier function in lamellar ichthyosis. To fully realize its therapeutic potential, delivery to target cells needs to be optimized, for example, through the use of tissue-specific promoters or capsid engineering. Improving transduction efficiency would enable a lower multiplicity of infection (MOI), potentially reducing the cytotoxicity and immunogenicity. While the vector successfully upregulated TGM1 expression in vivo and was associated with only minor, reversible changes in blood biochemistry, further investigations are needed to refine the methods for assessing its efficacy accurately.

Due to the fact that wild-type animals were used in the study, an objective assessment of the therapeutic potential of the proposed gene strategy requires experiments on model animal systems that mimic autosomal recessive congenital ichthyosis. It is important to take into account that the use of AAVs excludes the probability of integration of their genetic material into the host genome of target cells. To ensure long-term stability of gene expression in regenerating epidermis, it is necessary to consider the use of alternative vector systems. Such systems, such as integrating vectors, have the ability to selectively target stem cells in the basal layer of the epidermis. Lentiviruses, non-integrating HSV serotypes, and other approaches could be explored as potential candidates.

## 4. Materials and Methods

### 4.1. TGM1 Gene Construct Design and Codon Optimization

The OptimumGene algorithm was used to optimize the codon composition of the nucleotide sequence of the CDS mRNA of the *TGM1* gene, taking into account factors that could potentially affect gene expression levels. These factors included codon shift, GC composition, CpG dinucleotide content, mRNA secondary structure, tandem repeats, restriction sites interfering with cloning, premature polyadenylation sites, and additional minor ribosome binding sites. The nucleotide sequence of the CDS mRNA of the human *TGM1* gene (GeneBank #NM_000359.3, 2454 bp) was used as a matrix for codon optimization (*coTGM1*). Codon optimization, de novo synthesis of the nucleotide sequence of the CDS mRNA of the *TGM1* gene, and its cloning into the pAAV-MCS plasmid vector were performed by GenScript (Piscataway, NJ, USA). The correct assembly of the genetic construct was verified by restriction analysis and sequencing. After codon optimization, the CDS mRNA of co*TGM1* gene was cloned into pAAV-MCS plasmid vector (Agilent Technologies, Santa Clara, CA, USA) using *Eco*RI/*Bam*HI restriction site recombinase. The final construct contained the following elements: a CMV promoter/enhancer, β-globin intron, optimized TGM1 sequence, and a human growth hormone (hGH) polyadenylation signal. The construct was flanked by AAV2 inverted terminal repeats (ITRs). The reporter construct was plasmid pAAV-GFP carrying the gene of green fluorescent protein GFP.

### 4.2. Restriction Analysis of Plasmid Constructs

The restriction digestion of pAAV-TGM1 and pAAV-GFP was performed in 10 μL of reaction mixture using FastDigest *Eco*RI (#FD0274, Thermo Fisher Scientific Inc., Waltham, MA, USA) and *Bam*HI (#FD0054, Thermo Fisher Scientific Inc., USA) enzymes according to the methodology recommended by the manufacturer. pDNA was used in the amount 300 ng for the reaction. The reaction mixture was incubated in the thermostat for 15 min at 37 °C and analyzed by electrophoresis in 1% agarose gel stained with ethidium bromide.

### 4.3. Preparation of Recombinant AAV and Characterization of Viral Particles

#### 4.3.1. Obtaining and Concentrating Viral Particles

AAV Helper free system (Agilent Technologies, USA) was used to obtain recombinant vectors. AAV293 cells were transfected using polyethylenimine (#408727, Sigma-Aldrich, St. Louis, MO, USA). AAV293 cells were cultured at 37 °C in a humidified atmosphere with 5% CO_2_, at a cell monolayer density of less than 50%, in DMEM/F-12 complete medium (#C470p, PanEco, Moscow, Russia) to which were added 10% fetal bovine serum (#S181B, BioSera, Cholet, France) 2 mM L-glutamine (#F032, PanEco, Moscow, Russia) and 1× penicillin and streptomycin antibiotic mixture (#A065p, PanEco, Moscow, Russia) (referred to as complete DMEM/F-12 medium). Immediately before transfection, the monolayer density was 70–80%; for this purpose, 2.2 × 10^6^ cells were spread on a 10 cm Petri dish the day before transfection. The next day, co-transfection with three plasmids (pAAV-TGM1, pAAV-RC9, and pHelper) was performed as previously described [33]. Seventy-two hours after transfection, 0.5% (*v*/*v*) Triton X-100 was added to the cell suspension. The lysate was centrifuged in 250 mL beakers at 3000× *g*, 10 min, 4 °C. Purification was performed by centrifugation in an iodixanol density gradient 350,000× *g*, 1 h, +18 °C on a Beckman Optima-XPN ultracentrifuge. Then, fractions of 60% + ½ 40% were collected.

The sample was dialyzed in dialysis bags with a 100 kDa pore size (Spectrum, #0867140, Stamford, CT, USA) against 1× phosphate-salt buffer (PBS)/213 mM NaCl/0.001% Pluronic F-68 buffer at +4 °C and sterilized by filtration using syringe filters with a pore diameter of 0.22 μm. The virus titer was determined by real-time polymerase chain reaction (PCR-RT) as described further in Section 4.5.2.

#### 4.3.2. Transmission Electron Microscopy

AAV samples were diluted in formulating buffer (2.27× PBS, 350 mM NaCl, 11.35% sorbitol, 0.0023% Pluronic F-68) depending on the initial concentration (1:10 dilution was optimal for 1–5 × 10^14^, 1:10 dilution was optimal for 10^13^, 1:5 dilution was optimal for 10^12^, and no dilution was required for 10^12^; if the concentration was unknown, the assay was performed without dilution followed by visual assessment), and the 1:5 dilution was performed with a pipette by mixing 5 µL of sample with 20 µL of water in an Eppendorf tube and vortexing for 30 s, after which the solution was stored for 14 days at 2–8 °C. Next, a contrast solution was prepared: 0.1 g of uranyl acetate was dissolved in 9.9 mL of water, filtered through a 0.2 μm membrane, and stored at 4 °C for up to 1 year.

For sample preparation, carbon film copper grids were treated with air plasma (5 s in a PELCO easiGlow™ unit), then 10 µL of the sample was applied to Parafilm, incubated for 2 min, removed excess with filter paper, and washed with water. Contrasting was performed with 10 µL of 1% uranyl acetate (1 min incubation), washed with water, dried at room temperature, and placed in a storage box. The results were analyzed on an HT7700 transmission electron microscope (Hitachi, Tokyo, Japan) at 100 kV, analyzing the distribution of particles after preparation. The diameter of viral particles was measured using the ImageJ program (version 1.54g) (National Institutes of Health, Bethesda, MD, USA).

### 4.4. Transfection and Transduction of Cell Cultures

The cells used in this work are as described below. HEK293 (Human Embryonic Kidney 293 cells) (ATCC: CRL-11268) is an immortalized primary human embryonic kidney cell line containing modifications for efficient assembly of adeno-associated virus. HaCaT (RRID: CVCL_0038), an immortalized keratinocyte line derived from human skin epidermis, is the target cell in this study. Primary human fibroblasts (PHF) were isolated from human dermis using standard techniques. Human neuroblastoma SH-SY5Y cells (ATCC: CRL-2266) were used as a line with low endogenous TGM1 expression. Cells were cultured in complete DMEM/F-12 medium in an incubator at 37 °C in a humidified atmosphere containing 5% CO_2_.

For transfection of cells with pAAV-TGM1 and pAAV-GFP, we used branched polyethylenimine 25 kDa (PEI, #408727, Sigma-Aldrich, St. Louis, MO, USA), a solution of positively charged polymer that forms compact, stable, positively charged complexes with nucleic acid molecules.

Cells were transduced with AAV2-TGM1 and AAV2-GFP with a multiplicity of infection (MOI) of 1 million genomic copies of AAV per cell. For this purpose, cells were seeded on a 6-well culture plate at 50,000 cells per well in complete DMEM/F-12 medium. AAV2, protamine sulfate (to a final concentration of 10 μg/mL), and DMEM/F-12 without the addition of serum (1 mL) were mixed. After cell attachment, the culture medium was changed to the prepared mixture with AAV2. Cells were incubated with AAV2 for 24 h, after which the medium was changed to fresh complete DMEM/F-12 medium. Transgene expression analysis and collection of conditioned medium for cytokine profile analysis (Section 4.7) were performed 48 h after transfection and 96 h after transduction.

Genetic modification of HEK293, HaCaT, HPF, and SH-SY5Y cells using AAV-GFP is necessary to control the quality of cell transduction since the resulting transgene has a detectable fluorescence. Initial visual assessment of the genetic modification of cells was performed using an Axio Observer.Z1 microscope and Axio Vision Rel. 4.8 software (CarlZeiss, Oberkochen, Germany).

Transgene expression was also analyzed on a FACS Aria III flow cytometer (BD Biosciences, Milpitas, CA, USA). For this purpose, genetically modified HEK293 cells were stripped by trypsinization, resuspended in 500 μL of phosphate-salt buffer, and analyzed in the spectrum of FITC fluorophore (excitation wavelength = 495 nm, emission wavelength = 525 nm).

### 4.5. Analysis of TGM1 Expression and Enzymatic Activity

#### 4.5.1. Western Blot Analysis

Cells (1 × 10^6^ cells) were lysed in RIPA buffer containing a mixture of Halt™ Protease and Phosphatase Inhibitor Cocktail (#78440, Thermo Fisher Scientific Inc., Waltham, MA, USA). The concentration of total protein in the extracted samples was determined using Pierce BCA Protein Assay Kit (#23225, Thermo Fisher Scientific Inc., Waltham, MA, USA) according to the protocol recommended by the manufacturer. Results were measured at a wavelength of 562 nm on a Tecan infinite 200 pro instrument (Tecan Trading AG, Zurich, Switzerland).

The isolated proteins (in equal amounts) were separated by gradient step electrophoresis in 4–12% polyacrylamide gel with sodium dodecyl sulfate (SDS-PAGE) and transferred to 0.22 μm PVDF membrane (#WGPVDF22, Servicebio, Huangshi, China) using paper blotting filters (#G6007-50, Servicebio, Wuhan, China) and a semi-dry transfer system (BioRad Laboratories, Hercules, CA, USA).

Before antibody staining, membranes were blocked in EveryBlot Blocking Buffer (#12010020, BioRad, Hercules, CA, USA) for 5 min at room temperature. Membranes were stained for one hour at room temperature in PBS with Tween-20 (0.1%) (PBST), supplemented with 5% bovine serum albumin (BSA) (#9048-46-8, Acros Organics, Waltham, MA, USA), with primary polyclonal antibodies against TGM1 (1 μg/mL, #PAC256Hu01, Cloud-Clone, Katy, TX, USA). Membranes were then washed in PBST buffer and stained with horseradish peroxidase (HRP)-conjugated secondary antibodies (#170-5046, BioRad, Hercules, CA, USA) for 2 h at room temperature in PBST with BSA.

To normalize the amount of protein, membranes were stained with antibodies against β-actin conjugated to HRP (#A00730, GenScript, Piscataway, NJ, USA) for 2 h at room temperature. Proteins were visualized with Clarity™ Western ECL substrate (#1705061, BioRad Laboratories, Hercules, CA, USA) using a ChemiDoc XRS+ instrument (BioRad Laboratories, Hercules, CA, USA). Relative signal intensity levels were determined using Image LabTM, version 6.0.1 (Bio-Rad Laboratories, Hercules, CA, USA).

#### 4.5.2. Real-Time Polymerase Chain Reaction

Total RNA (tRNA) was isolated from cells using TRIzol reagent (#15596026, Invitrogen, Carlsbad, CA, USA) according to the manufacturer’s instructions. Organs were homogenized in 500–1000 μL of TRIzol reagent and sodium acetate buffer with the addition of glass beads in a FastPrep-24 homogenizer (MP Biomedicals, Irvine, CA, USA). The homogenizer was exposed for 20 s at a frequency of 5 Hz. After that, the homogenate was centrifuged for 5 min at 9500× *g*, and the supernatant was used for RNA isolation and determination of enzymatic activity according to the instructions recommended by the manufacturer (Section 4.5.4).

The concentration of total RNA samples was measured using a DS-11 FX+ spectrophotometer (DeNovix, Wilmington, DE, USA). cDNA synthesis was performed using the MMLV RT kit (#SK021, Eurogen, Moscow, Russia) according to the manufacturer’s instructions. PCR-RT was performed using specific primers and a probe to the nucleotide sequence of the TGM1 gene; primers and a fluorescent probe specific to the inverted terminal repeat (ITR) sequence of the AAV were used to estimate the virus titer. The primer sequences for TGM1 and ITR were designed using GenScript Online RT-PCR (TaqMan) Primer Design Tool software (GenScript, Piscataway, NJ, USA) and synthesized by Eurogen (Russia) (Table 1).

PCR amplification was performed on a CFX96 Touch™ RT-PCR Detection System (Bio-Rad, Hercules, CA, USA) with the following cycling mode: 95 °C—5 min, 35 cycles of denaturation at 95 °C—10 s, annealing at 57 °C for *coTGM1* sequence—30 s, elongation at 72 °C—30 s, 35 cycles. For the *18S* sequence, preheating at 95 °C—5 min, 35 cycles of denaturation at 95 °C—10 s, annealing at 50 °C—30 s, elongation at 72 °C—30 s 35 cycles. For *ITR* sequence—preheating at 95 °C—15 min, 39 cycles of denaturation at 95 °C—30 s, annealing at 61 °C.

The mRNA levels of the tested genes were normalized relative to the mRNA level of 18S ribosomal RNA. The relative level of mRNA transcription was calculated by comparative quantification of the threshold cycle value (ΔΔCT). For the in vitro samples, the absolute mRNA copy numbers were additionally normalized using the 18S normalization coefficient.

#### 4.5.3. Immunocytochemical Analysis

Cells were cultured in a 24-well plate on glass. The next day after seeding, transfection or transduction was performed. The cells were further cultured for another 4–6 days. The medium was removed, and the cells were fixed with 10% formalin (300 µL per well) for 25 min. The cells were washed three times with DPBS for 5 min. For intracellular staining, 300 μL per 1 well of 0.1% Triton X-100 was added. Incubation was performed for 20 min. Primary polyclonal antibody against TGM1 (#PAC256Hu01, Cloud-Clone, Houston, TX, USA) was diluted 0.5 μL per 500 μL PBS and incubated for 1.5 hr. Sample washed three times with DPBS for 5 min. Diluted secondary antibodies conjugated to Alexa Fluor™ 555 (#A-214281, Invitrogen, Carlsbad, CA, USA) 1 μL per 500 μL PBS and incubated for 1 h in the dark. Washed three times with DPBS for 5 min. The samples were encapsulated in Mounting Medium and visualized on an LSM 700 confocal microscope (Carl Zeiss, Oberkochen, Germany) using Zen black 2012 software (Carl Zeiss, Oberkochen, Germany). All samples were visualized using the same confocal settings (laser intensity, gain and offset). Semi-quantitative assessment of the arithmetic mean fluorescence intensity was performed as described further in Section 4.8.4).

#### 4.5.4. Evaluation of TGM1 Enzymatic Activity

TGM1 enzymatic activity was determined in lysates collected from native and genetically modified HEK293, HaCaT, and SH-SY5Y cells. Total protein concentration was determined using the Pierce BCA Protein Assay Kit (#23225, Thermo Fisher Scientific Inc., Waltham, MA, USA). Samples were normalized according to the concentration of total protein in the sample with the lowest concentration.

To determine the enzymatic activity of TGM1, a commercial TG1–CovTest kit (#opr0038, Covalab, France) was used according to the manufacturer’s instructions. Optical density was measured at 450 nm on a Tecan infinite 200 pro instrument (Tecan Trading AG, Zurich, Switzerland). To quantify TGM1 activity, a calibration curve was constructed using recombinant human TGM1 enzyme (50 μU/mL corresponded to an OD450 of 1.4 ± 0.06). All samples were analyzed in duplicate, monitoring nonspecific binding by negative control with ethylenediaminetetraacetic acid (EDTA).

### 4.6. Cell Viability Assessment

The viability of HEK293 cells at 3 and 7 days after AAV2-GFP viral transduction was determined using Alexa Fluor^®^488 Annexin V/Dead Cell Apoptosis Kit (#V13241, Thermo Fisher Scientific Inc., Waltham, MA, USA) according to the manufacturer’s instructions on a FACSAria instrument (Bioscience, Conshohocken, PA, USA) using BD FACSDivaTM software version 7.0.

### 4.7. Analysis of Cytokine Profile of Conditioned Cell Culture Medium

The Bio-Plex Pro Human Cytokine Screening Panel, 48-Plex kit (#12007283, Bio-Rad, USA) was used to evaluate the cytokine profile of conditioned medium of HaCaT and PHF cells collected 48 h after transfection and 96 h after transduction. The assay was performed according to the manufacturer’s instructions. The obtained data were analyzed using the MAGPIX instrument (Luminex, Austin, TX, USA) and xPONENT software (version 4.2.1324.0) (Luminex, Austin, TX, USA).

### 4.8. Working with Laboratory Animals

#### 4.8.1. Laboratory Animals, Groups, and Content

Ninety 12-week-old female and male Wistar line rats (weighing 230–250 g) were used in the study. The animals were randomly divided into three groups and three time points: control (5 animals per time point injected with 500 μL PBS), reference (5 animals per time point injected with AAV2-GFP), and experimental (5 animals per time point injected with AAV2-TGM1) at 7, 14, 21, days, respectively. Animals were kept in clear plastic cages, with a light:dark cycle of 12:12 h, at a temperature of 21–25 °C, and relative humidity of 40% to 70%, with unrestricted access to food and water. Scarification was performed using a scalpel; shallow wounds of 0.5 × 0.5 cm were made on the back of the rats. All experiments were conducted in accordance with ethical standards and current legislation, and the protocol was approved by the Animal Ethics Committee of Kazan Federal University (№ 50, approved 26 September 2024).

Recombinant AAV2-TGM1 was injected intradermally and applicatively at a concentration of 1 × 10^10^ per 1 kg in 300 μL PBS. Animals in the control group were injected with 100 μL of PBS. For application, the excipient gel was mixed with the viral vector in PBS at a ratio of 1.5:1. Before injection, whole blood samples were collected from rats in tubes containing the anticoagulant sodium citrate.

Laboratory pigs were also used to evaluate the functionality and safety of AAV2-TGM1. Animals were divided into 2 groups: control (5 individuals injected with PBS) and experimental (5 individuals injected with AAV2-TGM1). One week before sampling, pigs were injected intradermally with AAV2-TGM1 into a pre-prepared skin area, approximately 1.5 × 1.5 cm in size. The efficacy of the construct was assessed on days 1, 3, and 7. The obtained skin samples were further used to analyze TGM1 gene expression and secretion by PCR-RT (Section 4.5.2) and immunofluorescence analysis (Section 4.8.4). The scheme of experiments is presented in Appendix A.

#### 4.8.2. Determination of Leukocyte Formula and Biochemical Parameters of Blood

Blood was collected from rats and pigs to assess biochemical parameters, and leukocyte formula was additionally analyzed in rats. Blood was collected from rats at 7, 14, and 21 days after AAV2-TGM1 and AAV2-GFP administration. For evaluation of leukocyte formula and biochemical parameters, blood was collected from the tail vein in a volume of 1 mL. For leukocyte formula analysis, blood was collected in 1.5 mL microtubes with anticoagulant added, and whole blood was used. Lymphocytes, monocytes, and granulocytes were counted using an Abacus Junior Vet 5 hematology analyzer (Diatron, Budapest, Hungary). For biochemical analysis, blood was collected in tubes without anticoagulant, incubated for 40 min at room temperature, and then centrifuged at 4 °C, 2000× *g* for 20 min. The obtained serum was used to determine the levels of aspartate aminotransferase (AST), alanine aminotransferase (ALT), total bilirubin, and creatinine on a ChemWell 2900 biochemical analyzer (Awareness Technology, Palm City, FL, USA).

Blood was collected from the ear vein of pigs on days 7, 14, and 21 after AAV2-TGM1 injection. Blood in a volume of 1 mL was collected into tubes without anticoagulant, incubated at room temperature for 40 min, and then centrifuged at 4 °C, 2000× *g* for 20 min. Levels of AST, ALT, total bilirubin, and creatinine in serum were determined using a ChemWell 2900 biochemical analyzer (Awareness Technology, Palm City, FL, USA).

#### 4.8.3. Determination of Immune System Organ Cellularity in Rats

Determination of immune system organ cellularity was performed to investigate the immunotoxicity of AAV2-TGM1. For this purpose, at 7, 14, and 21 days after administration of AAV2-TGM1, AAV2-GFP, or PBS, rats were subjected to CO_2_ euthanasia; the thymus, lymph nodes, and tubular bones were extracted. Lymphoid organs were homogenized and resuspended in cooled Hanks’ solution (pH = 7.2–7.4) (#P020p, PanEco, Moscow, Russia) in a volume of 5 mL for thymus and in a volume of 1 mL for hamstring lymph nodes and bone marrow. The resulting suspension was filtered through two layers of capron and washed twice by centrifugation at 200× *g* for 5 min. Then, 10 μL of the suspension was added to 0.4 mL of 3% acetic acid solution with trypan blue; the concentration of nucleated cells in the organ was counted, and the ratio of living and dead cells was determined using Goryaev’s chamber.

#### 4.8.4. Immunofluorescence

Transversal paraffin skin sections were used to analyze TGM1 secretion in tissues. Sections were deparaffinized with solutions of xylene, alcohol, and distilled water and stained first with primary polyclonal antibodies against TGM1 (#PAC256Hu01, Cloud-Clone, USA) and then with secondary antibodies conjugated to Alexa Fluor™ 647 (#A-21244, Abcam, Waltham, MA, USA). Nuclei were stained with 4′,6-diamidino-2-phenylindole (DAPI) (#D8417, 10 μg/mL, Sigma-Aldrich, St. Louis, MO, USA). Coverslips were mounted on slides using ImmunoHistoMount medium (#ab104131, Abcam, Waltham, MA, USA). Sections incubated with secondary antibodies only (without primary antibodies) were used as reaction controls. The slides were examined and photographed with a confocal scanning microscope LSM 700 (Carl Zeiss, Oberkochen, Germany) using Zen black 2012 software (Carl Zeiss, Oberkochen, Germany).

Five sections in rats or pigs in the area of injection/application were analyzed in intact control (n = 5) and experimental group (n = 5) rats. The mean of fluorescence intensity (MFI units, semiquantitative analysis) was measured on confocal images of this area. Five areas in each region were examined, and for each channel, the lowest intensity signals were removed to minimize background. For semiquantitative analysis, all slices were visualized using identical confocal settings (laser intensity, gain, offset).

#### 4.8.5. Histopathological Analysis

On the 7th day after vector injection, experimental animals were euthanized. Fragments of the following organs were collected from each animal: skin, lungs, kidneys, heart, spleen, and liver. Each of the above fragments was placed in 10% formalin solution. After 48 h from the beginning of fixation, each fragment was embedded in paraffin; the obtained samples of internal organs were sliced on Minus S700A microtome (RWD, Thermo Fisher Scientific, Waltham, MA, USA) with a thickness of 5–7 μm, then the slices were horizontally dried on glass in a thermostat of 60 °C for 1 h, followed by a dewaxing step to remove paraffin before staining. The sections were stained with hematoxylin (BioVitrum, Saint Petersburg, Russia) and eosin (BioVitrum, Saint Petersburg, Russia) or Mallory’s trichrome (BioVitrum, Saint Petersburg, Russia). The stained sections were encapsulated in Consul-Mount mounting medium (#9990440, Thermo Fisher Scientific, Waltham, MA, USA) and examined using an APERIOCS2 light scanning microscope (Leica, Wetzlar, Germany).

Epidermal thickness was measured in control and experimental groups of animals using an electronic ruler in NPD.view software. For each sample, multiple measurements were performed in standardized areas of histological sections. The obtained data were used for statistical analysis of morphological changes in the epidermis.

### 4.9. Statistical Analysis

The data were analyzed using GraphPad Prism 8 software (GraphPad Software, Boston, MA, USA). The normality of data distribution was determined by the Anderson–Darling, D’Agostino–Pearson, Shapiro–Wilk, and Kolmogorov–Smirnov methods. Depending on the type of data distribution, analysis of variance (ANOVA) followed by Tukey’s post hoc test or the Kruskal–Wallis method was used for analyses with 3 or more comparison groups. Student’s t-test or the Mann–Whitney method was used for 2 comparison groups for analyses, respectively. Quantitative variables are presented through their mean and standard deviation. Statistically significant differences are labeled as *—*p* < 0.05, **—*p* < 0.01 and ***—*p* < 0.001, and ****—*p* < 0.0001.

## 5. Conclusions

The present study showed that recombinant AAV2-TGM1 provides *TGM1* expression in rat and pig skin after topical application and intradermal injection administration. Increased levels of *TGM1* mRNA and protein were confirmed by RT-PCR, immunofluorescence, and fluorescence intensity analysis. The histological structure of the skin, parenchymatous organs, and the CNS was preserved, and no signs of acute toxicity were detected.

Biochemical and immunological parameters demonstrate an acceptable safety profile: changes in bilirubin, creatinine, and cell viability were transient and were not accompanied by significant inflammation. These findings confirm the promising potential of AAV2-TGM1 for gene therapy of skin diseases. However, challenges such as high expression variability, limited duration of action, and the need for multiple administrations due to anti-vector immunity highlight the necessity for further optimization of the vector design, delivery method, and testing to *TGM1*-deficient animal models for a more objective assessment of therapeutic efficacy.

## Figures and Tables

**Figure 1 ijms-26-09976-f001:**
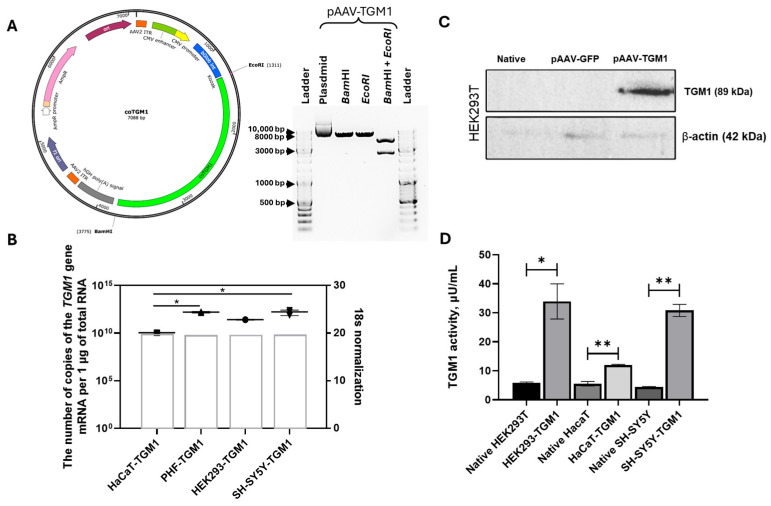
Functional analysis of the pAAV-TGM1 plasmid. (**A**) Map of the pAAV-TGM1 plasmid vector. (**B**) *TGM1* mRNA expression levels in native and pAAV-TGM1 transfected HEK293, HaCaT, SH-SY5Y, and PHF cells. Data were obtained by RT-PCR (48 h after transfection) and are presented as the mean of 3 biological repeats ± SD. Target gene expression levels were normalized to 18S rRNA. The gray box indicates the expression level of 18S represented in amplification cycles. (**C**) Western blot analysis of TGM1 (89 kDa) protein expression in lysates from native and transfected HEK293 cells (48 h after transfection). (**D**) TGM1 enzymatic activity in native and pAAV-TGM1 transfected HEK293, HaCaT, and SH-SY5Y cells. TGM1 enzymatic activity was measured using the TGI-COVEST kit (#opr0038, Covalab, Bron, France) 48 h after transfection. Data are presented as the mean of 3 biological replicates ± SD, compared to native cells. *—*p* < 0.05, **—*p* < 0.01.

**Figure 2 ijms-26-09976-f002:**
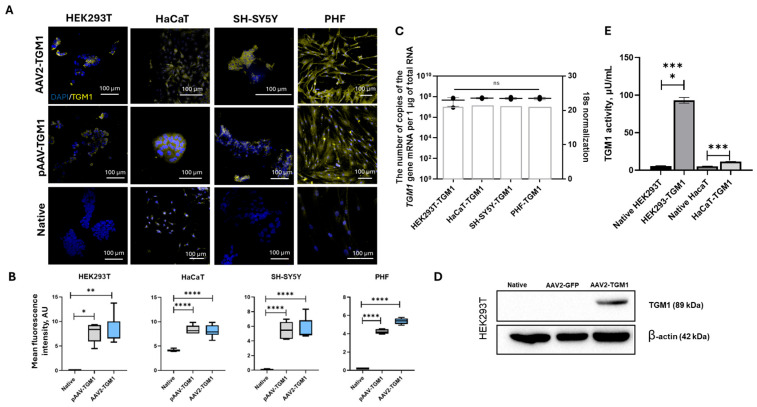
Functional analysis of recombinant AAV2-TGM1. (**A**) Immunofluorescence analysis of TGM1 protein expression in HEK293, SH-SY5Y, HaCaT, and PHF cells 96 h after AAV2-TGM1 transduction. (**B**) Analysis of the mean fluorescence intensity in modified cells. (**C**) Analysis of *TGM1* mRNA levels in AAV2-TGM1 transduced cells (96 h after transduction). Data were obtained by RT-PCR and are presented as the mean of 3 biological replicates ± SD. Target gene expression levels were normalized to 18S rRNA. The gray box indicates the expression level of 18S represented in amplification cycles. (**D**) Western blot analysis of TGM1 (89 kDa) and β-actin (42 kDa) protein expression in lysates from in native and transduced HEK293 cells. (**E**) TGM1 enzymatic activity in native and AAV2-TGM1 transduced HEK293 and HaCaT cells. *—*p* < 0.05, **—*p* < 0.01, ***—*p* < 0.001, ****—*p* < 0.0001, ns > 0.05.

**Figure 3 ijms-26-09976-f003:**
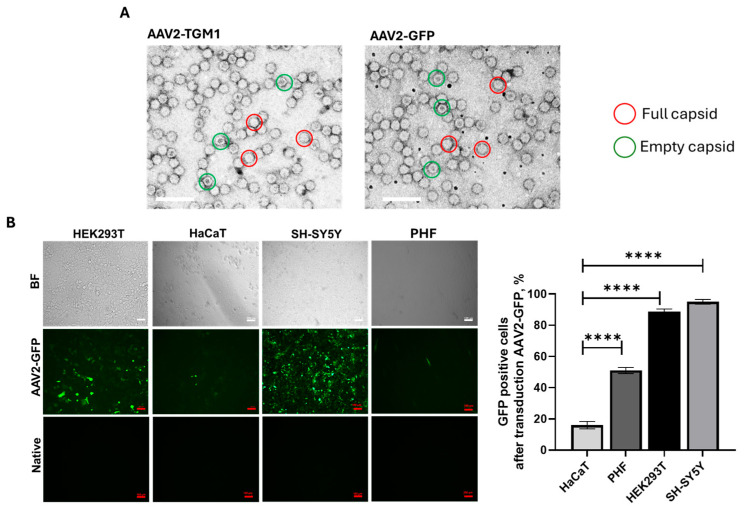
Evaluation of the functionality and safety of recombinant AAVs in vitro. (**A**) Transmission electron microscopy of recombinant AAV2-GFP and AAV2-TGM1 particles using negative staining. Images were at a magnification of 50,000×. The scale bar is 100 nm. (**B**) Analysis of AAV2-GFP transduction efficiency of HEK293, SH-SY5Y, HaCaT, and PHF cells by fluorescence microscopy. BF—brightfield. AAV2-GFP—cells after AAV2-GFP transduction. Native—native cells. Scale bar: 100 µm. ****—*p* < 0.0001.

**Figure 4 ijms-26-09976-f004:**
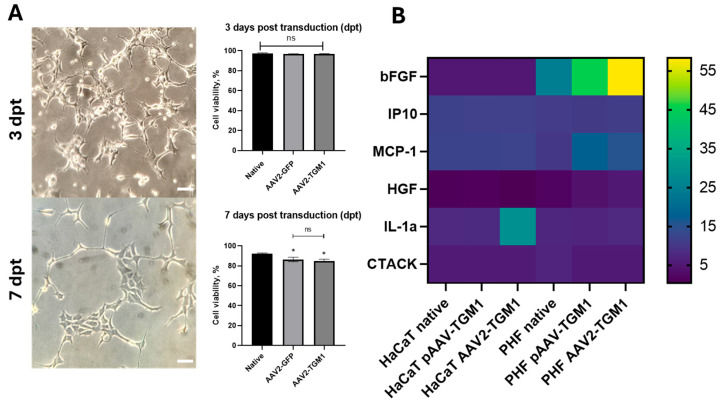
(**A**) Viability assessment of HEK293 cells transduced with AAV2-TGM1 and AAV2-GFP at 3 and 7 days post-transduction. Data were obtained by flow cytometry and are presented as the mean of 3 biological replicates ± SD. dpt—days after transduction. Scale bar: 20 µm. (**B**) Cytokine profile analysis of the conditioned medium of HaCaT and PHF cells after pAAV-TGM1 transfection and AAV2-TGM1 transduction. *—*p* < 0.05, ns > 0.05.

**Figure 5 ijms-26-09976-f005:**
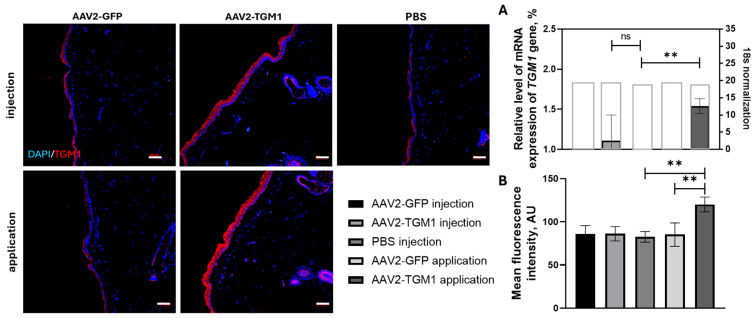
Evaluation of AAV2-TGM1 functionality after intradermal injection and topical application in rats. (**A**) Evaluation of the amount of *TGM1* gene mRNA in rat skin homogenates at 7 days after AAV2-TGM1 administration. Data were obtained by RT-PCR and are presented as the mean of 5 biological replicates ± SD. Target gene expression levels were normalized to 18S rRNA. The gray box indicates the expression level of 18S represented in amplification cycles. (**B**) Immunofluorescence analysis of TGM1 expression in rat skin by measuring mean fluorescence intensity (7 days after administration of AAV2-TGM1 or AAV2-GFP). Data are presented as the mean of 5 biological replicates. Scale bar: 50 μm. **—*p* < 0.01, ns > 0.05.

**Figure 6 ijms-26-09976-f006:**
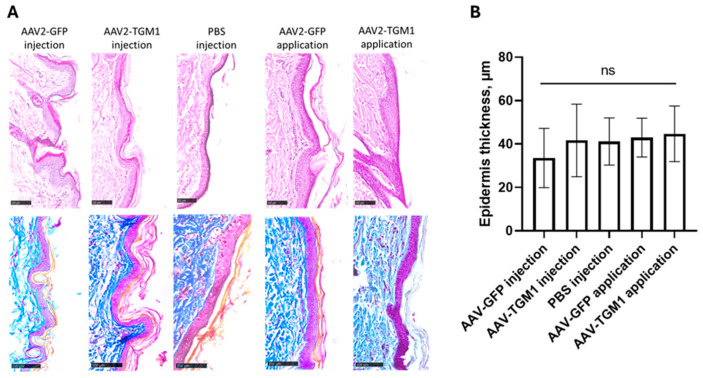
(**A**) Histopathological analysis of rat skin. Hematoxylin and eosin staining (**top row**) and Mallory’s trichrome staining (**bottom row**) magnification 20×. Experimental groups: AAV2-GFP (intradermal injection and topical application); AAV2-TGM1 (intradermal injection and topical application). Samples were analyzed 7 days after vector administration. Scale bar: 100 μm. (**B**) Measurement of epidermal thickness in experimental and control groups using a digital measuring tool in NPD.view software (version 2.9.29). Data are presented as the mean of 5 biological replicates ± SD, ns > 0.05.

**Figure 7 ijms-26-09976-f007:**
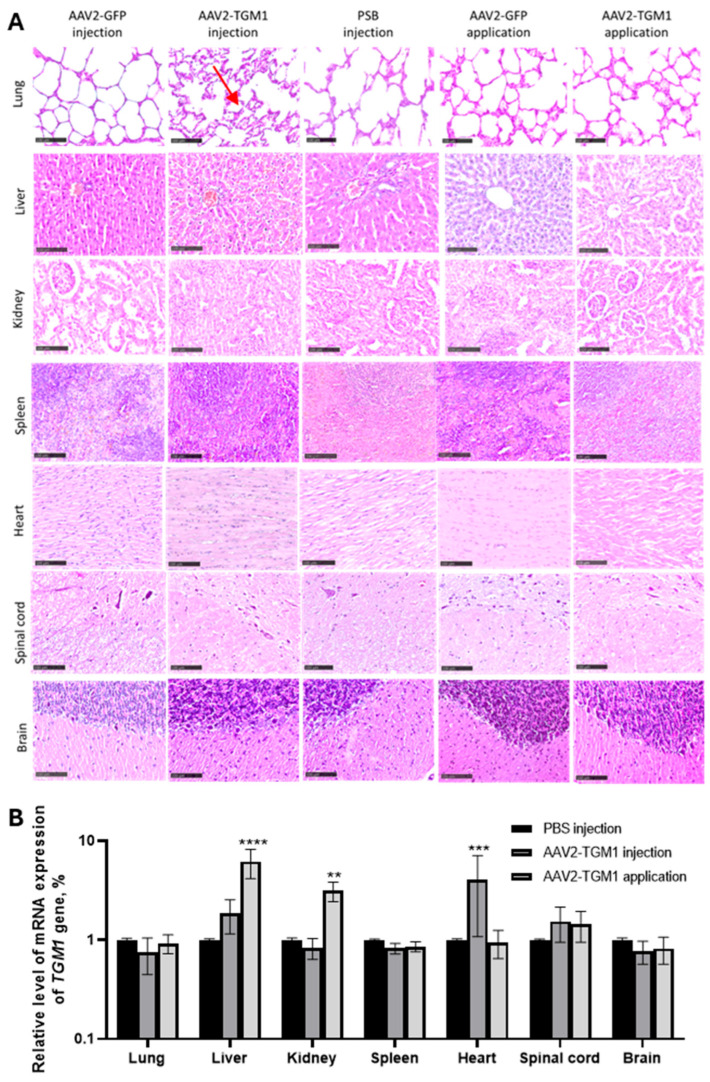
(**A**) Histopathological analysis of parenchymatous organs and the central nervous system of rats. Hematoxylin and eosin staining. Experimental groups: AAV2-GFP (intradermal injection and topical application); AAV2-TGM1 (intradermal injection and topical application). Samples were collected and analyzed 7 days after vector administration. Scale bar: 100 μm. (**B**) Analysis of AAV2-TGM1 biodistribution. Assessment of *TGM1* gene mRNA levels in rat organ homogenates 7 days after intradermal injection and topical application of AAV2-TGM1 using RT-PCR. **—*p* < 0.01, ***—*p* < 0.001, ****—*p* < 0.0001.

**Figure 8 ijms-26-09976-f008:**
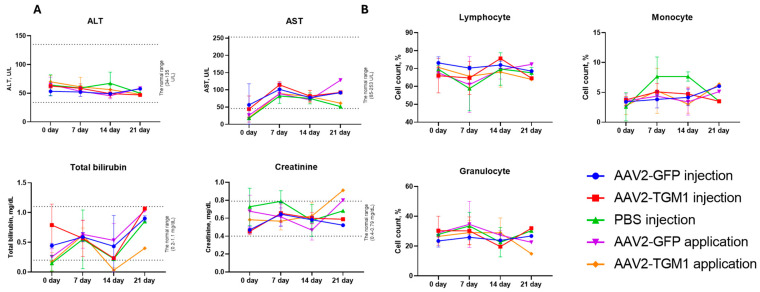
(**A**) Blood biochemical parameters in rats at 0, 7, 14, and 21 days after AAV administration. ALT—alanine aminotransferase, AST—aspartate aminotransferase. Data were obtained on a ChemWell 2900 biochemical analyzer (Awareness Technology, Palm City, FL, USA). (**B**) Leukocyte differential counts in rats at 0, 7, 14, and 21 days after AAV administration. Data were obtained on an Abacus Junior Vet 5 hematology analyzer (Diatron, Budapest, Hungary). Data are presented as the mean of 5 biological replicates ± SD.

**Figure 9 ijms-26-09976-f009:**
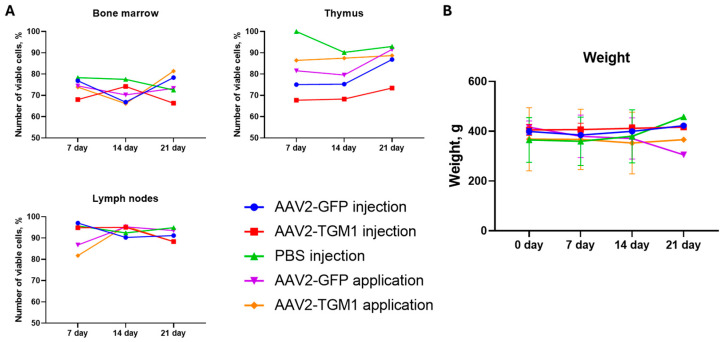
(**A**) Cell viability in immune organs at 7, 14, and 21 days after AAV administration. Viability was assessed by trypan blue exclusion assay. (**B**) Body weight of rats at 0, 7, 14, and 21 days after AAV administration. Data are presented as the mean of 5 biological replicates ± SD.

**Figure 10 ijms-26-09976-f010:**
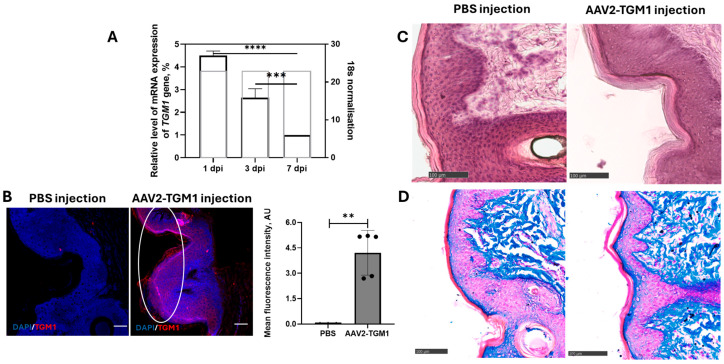
Evaluation of AAV2-TGM1 functionality upon intradermal injection to pigs. (**A**) Evaluation of the amount of TGM1 gene mRNA in pig skin homogenates at 1, 3, 7, days after AAV2-TGM1 injection. Data were obtained by RT-PCR and are presented as the mean of 5 biological replicates ± SD. dpi—days post-injection. Target gene expression levels were normalized to 18S rRNA. The gray box indicates the expression level of 18S represented in amplification cycles. (**B**) Immunofluorescence analysis of TGM1 expression in pig skin with evaluation of mean fluorescence intensity (7 days after AAV2-TGM1 injection). Data are presented as the mean of 5 biological repeats ± SD. Scale bar: 50 μm. (**C**) Histopathological analysis of pig skin, hematoxylin-eosin staining (7 days after AAV2-TGM1 injection). Scale bar: 100 μm. (**D**) Mallory’s trichrome staining (7 days after AAV2-TGM1 injection). **—*p* < 0.01, ***—*p* < 0.001, ****—*p* < 0.0001.

**Figure 11 ijms-26-09976-f011:**
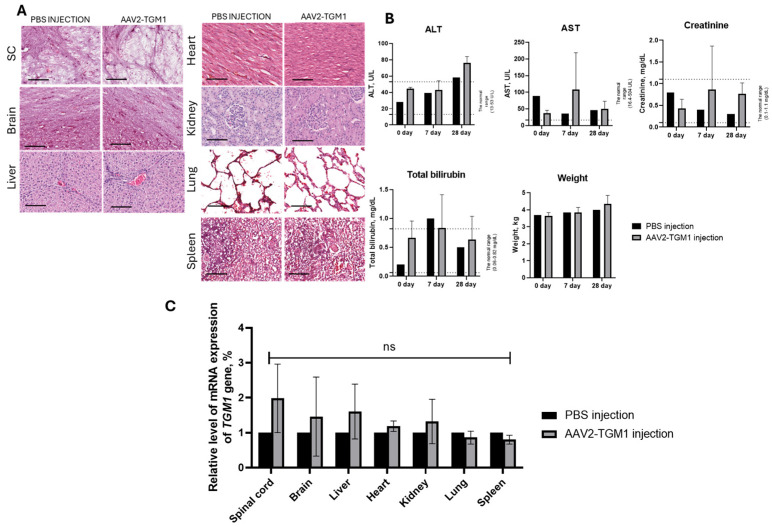
(**A**) Histopathological analysis of parenchymatous organs and the central nervous system of laboratory pigs. Hematoxylin and eosin staining. Experimental groups: PBS-injected control; AAV2-TGM1-injected group. Samples were analyzed 7 days after vector administration. Scale bar: 100 μm. (**B**) Evaluation of blood biochemical parameters in pigs at 0, 7, 14, and 21 days after vector administration. ALT—alanine aminotransferase and AST—aspartate aminotransferase are the indicators. Data were obtained using ChemWell 2900 biochemical analyzer (USA) and are presented as the mean of 5 biological replicates ± SD. (**C**) Analysis of AAV2-TGM1 biodistribution. Assessment of *TGM1* gene mRNA levels in pig organ homogenates 7 days after intradermal injection of AAV2-TGM1 using RT-PCR, ns > 0.05.

**Table 1 ijms-26-09976-t001:** Nucleotide sequences of primers and fluorescent probes.

Gene	Forward Primer (5′−3′)	Reverse Primer (5′−3′)	Fluorescent Probe (5′−3′)
ITR	GGAACCCCTAGTGATGGAGTT	CGGCCTCAGTGAGCGA	[FAM]CACTCCCTCTCTGCGCGCTCG[BBQ]
*coTGM1*	GACACCCCATTCATCTTTGC	TCTTAAAGCTGCCGTCATCC	[FAM]TGCCAGTACACCTTATCGCTGTTCAC[BBQ]
*18S*	GCGAGAAGATGACCCAGGATCGCCGCTAGAGGTGAAATTCTTG	CCAGTGGTACGGCCAGAGGCATTCTTGGCAAATGCTTTCG	[6-FAM]ACCGGCGCAAGACGGACCAG[BH2]

## Data Availability

The original contributions presented in this study are included in the article. Further inquiries can be directed to the corresponding author(s).

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
