# Peer review of "Evaluation of the Efficacy of Transglutaminase 1 Gene Delivery by Adeno-Associated Virus into Rat and Pig Skin and Safety of ARCI Gene Therapy"

_ijms, 2025, doi:10.3390/ijms26209976_

Round 1
Reviewer 1 Report
Comments and Suggestions for Authors
The authors present a typical approach to gene therapy using AAV; however, this work introduces a novel aspect compared to previously established methods, namely the topical administration of AAV. This route of administration is highly relevant to the study and requires robust evidence demonstrating that AAV can indeed function effectively when delivered topically. Consequently, the main concern of this work lies in clarifying how dermal cells are transduced by AAV when it is administered intradermally or topically.
In addition, while the toxicological analysis is of interest, it should also include an assessment of AAV presence to ensure a more complete evaluation.
Overall, several points need to be clarified or further developed in order to present a manuscript supported by sound and reliable results:
- The authors provide a detailed description of AAV production. However, the promoter used is not clearly specified. I assume it is the CMV promoter from the AAV-MCS construct, but this should be explicitly stated.
- Figure 1 requires improvement. In panel B, the background of the technique should be shown by including results obtained from cells transduced with an empty vector. Moreover, the results should be normalized against a housekeeping gene for proper comparison. This applies to all the mRNA quantifications in all the figures.
- Cytotoxicity assays in cultured cells should include results obtained with an appropriate control vector, such as an empty AAV or an AAV-GFP, in order to determine whether the observed effects are attributable to the vector itself or to the transgene.
- In the discussion, the rationale for topical administration is drawn from its successful use with HSV. However, no mechanistic explanation is provided for how this strategy would be effective with AAV. Specifically, it remains unclear how AAV could transduce epidermal and dermal cells, given that this vector does not readily traverse physical barriers.
- It is essential to characterize AAV transduction in rats by quantifying AAV genomes and transgene mRNA expression using PCR or ISH, as these represent the most reliable methods to assess the presence of AAV in tissues. This step is particularly critical when introducing a novel route of administration, such as topical delivery. Furthermore, quantification based solely on mean fluorescence intensity is not the most accurate approach for evaluating AAV transduction across groups, as it should also take into account the area of tissue analyzed.
- When assessing vector toxicity in different organs, it is also necessary to determine both the presence of the vector and transgene expression, especially when using a strong promoter such as CMV.
- Why was the intradermal route chosen in pigs if topical administration is the only approach shown to be effective in rats?
- It is unusual to detect transgene expression in pig skin as early as 24 hours after vector administration, with lower expression observed in the following days. This pattern contrasts with the typical kinetics reported in previous studies, where expression generally increases over time. A possible explanation could be contamination with AAV genomic DNA, which would be consistent with the observed results.
All of these issues need to be addressed in order to provide a robust and convincing demonstration of the proposed findings.
Author Response
The authors thank the Reviewer for thorough assessment of manuscript and their valuable comments. We sincerely apologize for the delay in our response, which was due to the time required to address the points raised comprehensively.
- The authors provide a detailed description of AAV production. However, the promoter used is not clearly specified. I assume it is the CMV promoter from the AAV-MCS construct, but this should be explicitly stated.
A text description of the plasmid has been added to the methodology in section 4.1. (line 544) and to the results (line 104). The key components of the final plasmid are schematically represented in Figure 1A.
- Figure 1 requires improvement. In panel B, the background of the technique should be shown by including results obtained from cells transduced with an empty vector. Moreover, the results should be normalized against a housekeeping gene for proper comparison. This applies to all the mRNA quantifications in all the figures.
The aim of this study was to evaluate the absolute number of mRNA transcripts in a cell after transduction/translation of the codon-optimized TGM1 sequence. In this method, the amount is estimated using a standard curve constructed from the dilution of the pAAV2-TGM1 plasmid. PCR using this method has been repeatedly validated in our previous studies [1–3].
The results of PCR for TGM1 after transduction with an empty vector will not be representative, since the primers are selected for the codon-optimized sequence and the values in the control (vehicle) groups will be threshold signal/0. That is why the graphs do not show control groups without transfection/transduction. If the task is to show the presence of the vector in the cells, Figure 3B demonstrates visually and using a cytometer the presence of the vector in the analyzed cells.
We normalized the results for 18S in in vitro experiments as follows: 18S expression levels in all experimental groups were normalized to the lowest value (the highest cycle). The obtained coefficients were multiplied by the absolute value of TGM1 mRNA copies in the experimental groups. This minimised differences in absolute values depending on PCR sample preparation (amount of RNA, varying quality/degradation, inhibitors in the sample, or the effect of transfection).
- Cytotoxicity assays in cultured cells should include results obtained with an appropriate control vector, such as an empty AAV or an AAV-GFP, in order to determine whether the observed effects are attributable to the vector itself or to the transgene.
The results of the cytotoxicity of the AAV2-GFP vector have been added to the article. To avoid overcluttering the article, the dot plot graphs from the cytometer have been removed.
- In the discussion, the rationale for topical administration is drawn from its successful use with HSV. However, no mechanistic explanation is provided for how this strategy would be effective with AAV. Specifically, it remains unclear how AAV could transduce epidermal and dermal cells, given that this vector does not readily traverse physical barriers.
Indeed, AAV encounters physical barriers in the skin, which limits its penetration into the epidermis and dermis, unlike the natural internalisation of HSV. In addition, Freedman's study, which evaluated HSV, also used a model of epidermal integrity disruption using acetone or tape strips. There are data in the literature on the possibility of skin cell transduction with local application of AAV. In particular, it has been shown that: microdamage or wounds increase the permeability of the epidermal barrier, facilitating vector access to basal keratinocytes and dermal fibroblasts [4], with differences in the transduction efficiency of different serotypes.
We consider topical application of AAV not as a universal strategy for all tissues, but specifically as an approach justified in conditions of skin pathologies where the barrier function may be impaired and where local intradermal injection or topical application, but on a scarification model, allows a therapeutic effect to be achieved through direct contact with target cells. Further studies are needed to assess the ability of AAV to penetrate normal, undamaged skin.
The text has been added to the discussion section (line 456).
- It is essential to characterize AAV transduction in rats by quantifying AAV genomes and transgene mRNA expression using PCR or ISH, as these represent the most reliable methods to assess the presence of AAV in tissues. This step is particularly critical when introducing a novel route of administration, such as topical delivery. Furthermore, quantification based solely on mean fluorescence intensity is not the most accurate approach for evaluating AAV transduction across groups, as it should also take into account the area of tissue analyzed.
MIF was calculated as follows: the average fluorescence intensity was assessed in five areas of the epidermis (one sample) in five samples in five biological replicates. Only the epidermal area was selected, and MIF was normalized to a single area value using ZEISS ZEN + ImageJ software.
The results of PCR in rats have been added to the article, which also explain the use of intradermal injection in the pig experiment.
- When assessing vector toxicity in different organs, it is also necessary to determine both the presence of the vector and transgene expression, especially when using a strong promoter such as CMV.
- Why was the intradermal route chosen in pigs if topical administration is the only approach shown to be effective in rats?
Pig skin is histologically and immunologically much closer to human skin than rodent skin: it is thicker and has a well-developed dermal layer. Given the limited penetration of AAV through the epidermal barrier, intradermal administration was chosen in the pig model, which is the standard delivery method and provides more predictable transduction, given the smaller sample size.
As shown in Figure 5A, expression was observed in the AAV2-TGM1 injection group, but there was no statistically significant difference from the control group. We planned to evaluate both injection and application in pigs, but due to delays in the delivery of animals, we only conducted the first part of the experiments. The second part (application) is planned and will be presented in a subsequent article.
- It is unusual to detect transgene expression in pig skin as early as 24 hours after vector administration, with lower expression observed in the following days. This pattern contrasts with the typical kinetics reported in previous studies, where expression generally increases over time. A possible explanation could be contamination with AAV genomic DNA, which would be consistent with the observed results.
Indeed, early detection of transgene expression in pig skin 24 hours after AAV administration differs from the typical kinetics described earlier, where expression levels typically increase over several days. We agree that one possible reason may be the presence of residual vector DNA, which in some cases can be detected in highly sensitive analyses. At the same time, we consider it necessary to note a number of additional factors that may explain the results obtained.
- It should be noted that by day 7, the amount of mRNA decreases significantly, which may be due to the processes of epidermal cell desquamation. Perhaps the level of 106 copies is not the peak expression, but since the time points were 1, 3, and 7, we could not detect the peak.
- It is possible that the recorded level reflects a short-term surge in episomal DNA activity.
- The use of non-integrating vectors such as AAV or non-replicative HSV-1 is transient in nature. Their expression decreases, partly due to the desquamation of epidermal cells [6]. (It was shown on Vyjuvek, Imlygic clinical trials)
We used archived samples and corrected the expression for 1 day. We assume that this was indeed contamination of the vector DNA and excluded this sample from the analysis/results. However, in our study, we did observe transgenes at day 1, although not at such high levels with transient expression. Upon normalisation, a 4.5-fold increase was observed at 1 day and a 2.16-fold increase at 3 days compared to 7 days. Normalisation is shown by the grey columns in the graphs. Sample preparation was performed correctly, with DNase treatment before RNA isolation.
- Shaimardanova, A.A.; Chulpanova, D.S.; Solovyeva, V.V.; Issa, S.S.; Mullagulova, A.I.; Titova, A.A.; Mukhamedshina, Y.O.; Timofeeva, A.V.; Aimaletdinov, A.M.; Nigmetzyanov, I.R.; et al. Increasing β-Hexosaminidase A Activity Using Genetically Modified Mesenchymal Stem Cells. Neural Regen. Res. 2023, 19, 212–219, doi:10.4103/1673-5374.375328.
- Mullagulova, A.; Shaimardanova, A.; Solovyeva, V.; Mukhamedshina, Y.; Chulpanova, D.; Kostennikov, A.; Issa, S.; Rizvanov, A. Safety and Efficacy of Intravenous and Intrathecal Delivery of AAV9-Mediated ARSA in Minipigs. Int. J. Mol. Sci. 2023, 24, 9204, doi:10.3390/ijms24119204.
- Shaimardanova, A.A.; Chulpanova, D.S.; Solovyeva, V.V.; Issa, S.S.; Mullagulova, A.I.; Titova, A.A.; Mukhamedshina, Y.O.; Timofeeva, A.V.; Aimaletdinov, A.M.; Nigmetzyanov, I.R.; et al. Increasing β-Hexosaminidase A Activity Using Genetically Modified Mesenchymal Stem Cells. Neural Regen. Res. 2024, 19, 212–219, doi:10.4103/1673-5374.375328.
- Keswani, S.G.; Balaji, S.; Le, L.; Leung, A.; Lim, F.-Y.; Habli, M.; Jones, H.N.; Wilson, J.M.; Crombleholme, T.M. Pseudotyped Adeno-Associated Viral Vector Tropism and Transduction Efficiencies in Murine Wound Healing. Wound Repair Regen. Off. Publ. Wound Heal. Soc. Eur. Tissue Repair Soc. 2012, 20, 592–600, doi:10.1111/j.1524-475X.2012.00810.x.
- Keswani, S.G.; Balaji, S.; Le, L.; Leung, A.; Lim, F.-Y.; Habli, M.; Jones, H.N.; Wilson, J.M.; Crombleholme, T.M. Pseudotyped Adeno-Associated Viral Vector Tropism and Transduction Efficiencies in Murine Wound Healing. Wound Repair Regen. Off. Publ. Wound Heal. Soc. Eur. Tissue Repair Soc. 2012, 20, 592–600, doi:10.1111/j.1524-475X.2012.00810.x.
- Sandoval, A.G.W.; Badiavas, E.V. Towards Extracellular Vesicles in the Treatment of Epidermolysis Bullosa. Bioengineering 2025, 12, 574, doi:10.3390/bioengineering12060574.
Reviewer 2 Report
Comments and Suggestions for Authors
In this study, as the first step of the gene therapy for lamellar ichthyosis (LI)/autosomal recessive congenital ichthyosis (ARCI) caused by mutations in gene for transglutaminase 1 (TGM1), the authors conducted extensive in vitro cell culture and in vivo animal studies using adeno-associated viral vector of serotype 2 containing TGM1 gene (AAV2-TGM1). The results suggested the usefulness and safety of this therapy for the future application in human patients. This is an comprehensive and well designed study for the functionality and safety for AAV2-TGM1 therapy, and the results were in general clear. This study provides several implications into the future gene therapy for LI/ARCI. However, I have several comments, which are described below.
- The name of the disease, LI or ARCI, may be better to be added in the title of this study for the easier understanding of the readers.
- The abbreviation of LI is used erroneously in the abstract section, which should be corrected.
- The authors used the different terms, i.e., intradermal and applicative injections and topical application, for the administration of the AAV2 for rats and pigs. However, these terms are confusing in some places, such as at lines 206-212 and lines 707, in the figure legend for the figure 5, and in the figure of appendix A. The authors should clearly mention about the differences of these distinct administration methods and the difference in each of different experiments in this study.
- In the section for histopathological findings for both rat and pig experiments, the authors may describe the difference of the findings in the epidermis between AAV2-TGM1 applied animals and controls animals, because the AAV2-TGM1 application may induce some changes in the epidermis.
- The vector seems to be incorporated into keratinocytes, although the authors used the fibroblast cells in the cell culture section. The authors may explain why they selected fibroblast as one of the cell lines.
- English may need some improvements in some places. This manuscript may be correct by native English speakers.
- In the sentence for the western blot analysis at lines 150-152, the term “an immunoprecipitation band” may be wrong.
- The full spell for the abbreviation “PHF” may be shown at line 108.
- The conclusions section may be moved before the materials and methods section.
Comments on the Quality of English Language
- English may need some improvements in some places. This manuscript may be correct by native English speakers.
Author Response
The authors thank the Reviewer for thorough assessment of manuscript and valuable comments. We sincerely apologize for the delay in our response, which was due to the time required to address the points raised comprehensively.
- The name of the disease, LI or ARCI, may be better to be added in the title of this study for the easier understanding of the readers.
The title has been corrected.
- The abbreviation of LI is used erroneously in the abstract section, which should be corrected.
Corrected.
- The authors used the different terms, i.e., intradermal and applicative injections and topical application, for the administration of the AAV2 for rats and pigs. However, these terms are confusing in some places, such as at lines 206-212 and lines 707, in the figure legend for the figure 5, and in the figure of appendix A. The authors should clearly mention about the differences of these distinct administration methods and the difference in each of different experiments in this study.
The text has been corrected: the term “intradermal injection” is used for AAV injection, the term “topical application” is used for gel application, and the term “administration” is used when describing two methods at once, for example, 7 days after vector use or when describing vector application in general.
- In the section for histopathological findings for both rat and pig experiments, the authors may describe the difference of the findings in the epidermis between AAV2-TGM1 applied animals and controls animals, because the AAV2-TGM1 application may induce some changes in the epidermis.
In our study, we did not find any changes in the epidermis. We added a mention of the possible effects of AAV use described in the literature to the discussion section (line 495).
- The vector seems to be incorporated into keratinocytes, although the authors used the fibroblast cells in the cell culture section. The authors may explain why they selected fibroblast as one of the cell lines.
The assessment of expression in fibroblasts is valuable for research into skin gene therapy in general. Therefore, the authors included these results, which they believe are of interest in terms of evaluating the effectiveness of AAV2-TGM1 in increasing expression in fibroblasts and comparing them with other cells. There are a number of diseases that fall into the group of hereditary ichthyoses associated with genetic abnormalities in fibroblasts, such as epidermolysis bullosa.
In addition, studies have shown that in patients with severe forms of congenital ichthyosis, fibroblasts demonstrate changes in the synthesis of extracellular matrix components and impaired secretion of growth factors and cytokines, which may affect skin regeneration and barrier function [1,2]. Thus, studies using fibroblasts may be useful for diseases of the dermal layer of the skin.
- English may need some improvements in some places. This manuscript may be correct by native English speakers.
The English has been carefully checked and corrected. Changes to the text are highlighted in red.
- In the sentence for the western blot analysis at lines 150-152, the term “an immunoprecipitation band” may be wrong.
The term has been corrected.
- The full spell for the abbreviation “PHF” may be shown at line 108.
Corrected
- The conclusions section may be moved before the materials and methods section.
authors used the IJMS template file. If necessary, the section can be moved higher.
- Chacón-Solano, E.; León, C.; Díaz, F.; García-García, F.; García, M.; Escámez, M.J.; Guerrero-Aspizua, S.; Conti, C.J.; Mencía, Á.; Martínez-Santamaría, L.; et al. Fibroblast Activation and Abnormal Extracellular Matrix Remodelling as Common Hallmarks in Three Cancer-Prone Genodermatoses. Br. J. Dermatol. 2019, 181, 512–522, doi:10.1111/bjd.17698.
- Fleckman, P.; Hager, B.; Dale, B.A. Harlequin Ichthyosis Keratinocytes in Lifted Culture Differentiate Poorly by Morphologic and Biochemical Criteria. J. Invest. Dermatol. 1997, 109, 36–38, doi:10.1111/1523-1747.ep12276450.
Round 2
Reviewer 1 Report
Comments and Suggestions for Authors
The authors have improved the manuscript by reanalyzing several experiments and reformulating multiple aspects. Although I still have some doubts about certain points, the manuscript appears to have sufficient significance to merit publication in this journal.